# *Mimosa tenuiflora* Aqueous Extract: Role of Condensed Tannins in Anti-Aflatoxin B1 Activity in *Aspergillus flavus*

**DOI:** 10.3390/toxins13060391

**Published:** 2021-05-29

**Authors:** Christopher Hernandez, Laura Cadenillas, Anwar El Maghubi, Isaura Caceres, Vanessa Durrieu, Céline Mathieu, Jean-Denis Bailly

**Affiliations:** 1Toxalim (Research Center in Food Toxicology), Université de Toulouse, INRAE, ENVT, EI-Purpan, 313000 Toulouse, France; hernandezhernandezchristopher@gmail.com (C.H.); laura.cadenillas.s@gmail.com (L.C.); anwar.vet2002@gmail.com (A.E.M.); isauracaceres@hotmail.com (I.C.); 2Laboratoire de Chimie Agro-Industrielle (LCA), Université de Toulouse, INRA, INPT, 4 Allée Emile Monso, 31030 Toulouse, France; vanessa.durrieu@ensiacet.fr (V.D.); celine.mathieu@toulouse-inp.fr (C.M.); 3Centre d’Application et de Traitement des Agro-Ressources (CATAR), INPT, Toulouse, 4 Allée Emile Monso, 31030 Toulouse, France

**Keywords:** *Aspergillus flavus*, *Mimosa tenuiflora*, aqueous extract, aflatoxin B1, inhibition, gene expression, condensed tannins

## Abstract

Aflatoxin B1 (AFB1) is a potent carcinogenic mycotoxin that contaminates numerous crops pre- and post-harvest. To protect foods and feeds from such toxins without resorting to pesticides, the use of plant extracts has been increasingly studied. The most interesting candidate plants are those with strong antioxidative activity because oxidation reactions may interfere with AFB1 production. The present study investigates how an aqueous extract of *Mimosa tenuiflora* bark affects both the growth of *Aspergillus flavus* and AFB1 production. The results reveal a dose-dependent inhibition of toxin synthesis with no impact on fungal growth. AFB1 inhibition is related to a down-modulation of the cluster genes of the biosynthetic pathway and especially to the two internal regulators *aflR* and *aflS*. Its strong anti-oxidative activity also allows the aqueous extract to modulate the expression of genes involved in fungal oxidative-stress response, such as *msnA, mtfA*, *atfA*, or *sod1*. Finally, a bio-guided fractionation of the aqueous extract demonstrates that condensed tannins play a major role in the anti-aflatoxin activity of *Mimosa tenuiflora* bark.

## 1. Introduction

Aflatoxins are toxic secondary metabolites produced naturally by various species of the *Aspergillus* genus and in particular by those belonging to the section *Flavi*. *Aspergillus flavus* and *Aspergillus parasiticus* attract the most attention of these toxigenic species because they are found in numerous crops where they can produce aflatoxins [1,2]. At least 20 different aflatoxin molecules have been identified, four of them being naturally produced by fungi (AFB1, -B2, -G1, and -G2) and the rest corresponding to metabolites that appear during liver metabolization of the fungal toxins in mammals [3,4].

Aflatoxin B1 (AFB1) is the most important member of this family. This compound is a potent natural carcinogen, classified as a group 1 carcinogenic compound for humans by the International Agency for Research on Cancer [5]. Its main pathogenic route is the induction of hepatocellular carcinoma, a type of cancer with a high mortality rate. This molecule is also teratogenic, genotoxic, and cytotoxic for many types of cells [6,7]. AFB1 is immune-toxic [8], and exposure to it has been closely related to impaired growth in children, including stunting and wasting [9]. AFB1 contamination is a major problem in tropical and subtropical regions where environmental conditions (temperature and humidity) are optimal for fungal growth and toxin production on crops both pre- and post-harvest. In these regions, seeds (peanuts), cereals (maize, rice), and spices are frequently contaminated with aflatoxins [10,11]. Moreover, global climate change has recently led to the appearance of this toxin in regions where it previously was not found, such as southern and central Europe, and it is feared that AFB1 contamination will continue to increase in these regions in coming years [12,13]. 

Different strategies are available to reduce AFB1 contamination, ranging from pre-harvest prevention (good agricultural and manufacturing practices) to post-harvest prevention (adequate storage practices and chemical detoxification) [14]. However, according to worldwide surveys, such measures do not completely eliminate AFB1 contamination [15]. Moreover, once AFB1 is produced, it is very difficult to remove it from food commodities or degrade it because it is highly stable, resistant to temperature extremes, and unaffected by most food processing [14,16]. It is therefore vital to find new strategies to limit AFB1 contamination of crops and subsequent exposure of consumers to this carcinogenic agent.

Different alternative strategies are developed to limit AFB1 contamination of foods and feeds. Among them, the use of atoxigenic fungal strains to compete with toxigenic ones is more and more used and different atoxigenic strains or mixtures of strains are now commercially available [17].

The use of natural compounds from plants has also been proposed as a promising alternative strategy to control AFB1 contamination [18]. Numerous studies have shown that extracts of certain herbs, spices, and plants display antifungal and anti-aflatoxigenic activity [19,20,21]. Plants are one of the richest sources of bioactive compounds, which serve not only to communicate between plants and to protect plants against aggressors such as insects, bacteria, and fungi, but also to combat various external stresses [22]. Polyphenols, alkaloids, and terpenes are three examples of plant secondary metabolites that hinder the production of AFB1 [21]. Many of these bioactive compounds exhibit a high anti-oxidant capacity that is linked to their primary function in the plant [23].

*Mimosa tenuiflora* (Willd.) Poir. (*Fabaceae* family and *Mimosoideae* subfamily) is a plant endemic of Mexico, known under the common name of “Tepezcohuite” [24,25]. Its bark is widely used in traditional Mexican and Brazilian medicine as an effective remedy to treat skin burns and wounds [24,26]. This plant may be used to reduce venous ulceration [27]; accelerate the healing process [28]; act as an antimicrobial agent against *Staphylococcus aureus* [29], *Micrococcus luteus*, *Bacillus subtilis*, and *Candida albicans* [25]; and regenerate bone [30]. Several studies have isolated and identified the compounds present in different parts of *M. tenuiflora*. Various flavonoids such as sakuranetin (5,4′-dihydroxy-7-methoxyflavanone), genkwanin (5,4′-dihydroxy-7-methoxyflavone), and sorbifolin (5,6,4′-trihydroxy-7-methoxyflavone) have been identified in the leaves of the plant [31]. Phytoindoles such as yuremamin [32], in addition to saponins and tannins, have been identified in the aqueous extract of *M. tenuiflora* bark [27]. Given that some of the active compounds identified from *M. tenuiflora* are possible antiradical molecules [33] and that oxidative stress is linked to aflatoxin synthesis [34], the present study evaluates the anti-aflatoxigenic activity of aqueous extract of *M. tenuiflora*. 

We thus investigate how *M. tenuiflora* stem bark Aqueous Extract (MAE) affects the growth and AFB1 production of *Aspergillus flavus*. The molecular mechanism is investigated by analyzing how MAE affects the expression of several genes from the AFB1 gene cluster and how it affects the gene network involved in both AFB1 regulation and the oxidative stress response in *A. flavus*.

Bio-fractionation was used to identify the active compounds responsible for this effect. The results indicate that the impact on toxin production correlates with total polyphenols, condensed tannins, and the anti-oxidant activity of each fraction.

## 2. Results and Discussion

### 2.1. Effect of M. tenuiflora Stem Bark Aqueous Extract on Fungal Growth and Aflatoxin B1 Production

#### 2.1.1. Impact on Fungal Growth

Five different concentrations of the MAE [0.025, 0.05, 0.10, 0.15, and 0.20 mg dry matter (DM) per mL] were analyzed and compared with a control culture without an extract to determine how they affect fungal growth. Eight days of incubation in concentrations below 0.15 mg DM/mL of the MAE has no statistically significant impact on the growth of *A. flavus* NRRL 62477. The same incubation period in 0.15 and 0.20 mg DM/mL produces a mild but statistically significant difference in the growth of *A. flavus* NRRL 62477, with a 3% reduction in the growth occurring at the higher concentration (Figure 1).

#### 2.1.2. Reduced Production of Aflatoxin B1

Figure 1 shows how the increasing concentration of MAE affects AFB1 production. MAE produces a dose-dependent reduction of AFB1 production in *A. flavus* NRRL 62477. Control cultures of *A. flavus* NRRL 62477 strain led to the production of 5.82 ± 0.48 µg AFB1/mL of culture medium after 8 days at 27 °C. This production was reduced to 2.87 ± 0.16 (reduction of 50%) and 2 ± 0.1 µg/mL (reduction of 66%) when the fungus was exposed to 0.15 and 0.20 mg DM/mL of MAE respectively. This reduction in AFB1 production exceeds that reported for other plant extracts such as essential oils of *Cuminum cyminum* and *Coriandrum sativum*, or aqueous extract of *Micromeria graeca*, which have a half-maximal inhibitory concentration (IC50) of 0.6 to 1.2 mg DM/mL [19,35]. 

#### 2.1.3. MAE Inhibition of Genes from AFB1 Cluster

AFB1 is the final product of an enzymatic cascade involving more than 20 reactions governed by 27 genes, which are grouped into a cluster and mainly regulated by the transcription factor *aflR* and its co-activator *aflS*. To study how MAE affects *A. flavus*, we evaluated the expression of several genes belonging to the AFB1 cluster. Figure 2 shows that MAE inhibits the expression of AFB1 cluster genes. In fact, the two main internal regulators are down-modulated with fold changes of 1.4 for *aflR* and 1.6 for *aflS* compared with the control. Moreover, gene coding for stable AF-intermediates intervening at the first (*aflC, aflD*), middle (*aflL*), and late (*aflO*, *aflP*) stages of the enzymatic cascade is also hindered by MAE. In contrast, *aflT* is not significantly affected by MAE. This result is consistent with the fact that this gene is not regulated by *aflR* nor *aflS* [36]. 

The inhibition of *aflR* and *aflS* expression induced by MAE is sufficient to repress the expression of the other genes of the cluster, thereby reducing AFB1 production. Indeed, normal expression of the *aflR* and *aflS* genes is essential to maintain the protein dimer formation necessary to activate most of the genes of the AFB1 cluster [37]. 

To date, few reports have discussed the use of aqueous extracts as AFB1 inhibitors, so their precise molecular mechanism remains unknown. Nevertheless, several studies have demonstrated that aqueous extracts from plants such as Puer tea or *Micromaeria graeca* reduce the AFB1 production of *A. flavus* without significantly affecting fungal growth [19,38]. In these studies, the expressions of the principal aflatoxin gene regulators *aflR* and *aflS* were also downregulated, which is consistent with the present results. 

#### 2.1.4. Effect of MAE on Expression of Genes Involved in Oxidative Stress Response 

Aflatoxin regulation is a complex process involving an interconnection of several gene networks that are modulated by environmental conditions. In *Aspergilli*, the synthesis of secondary metabolites including AFB1 is closely related with the oxidative stress response [39]. For instance, in *A. flavus* and *A. parasiticus*, a regulatory network of several bZIP transcription factors has been demonstrated to regulate both AFB1 biosynthesis and the oxidative stress response [40,41]. 

As shown below, MAE produces a strong antiradical activity. Thus, to better understand how it affects the oxidative stress response of *A. flavus*, we analyze how it affects a stress-related gene network. For this purpose, we study gene coding for catalases and superoxide dismutase synthesis (*catA*, *cat2*, *sod1*, *mnSOD*); the global transcription factor *mtfA*; several bZIP transcription factors (*atfA, atfB, ap-1*); and the genes *srrA* and *msnA*, which participate in the fungal stress response signaling pathway. 

As shown in Figure 3, several genes are significantly upregulated upon adding MAE. This includes the conidia-catalase gene *catA*, which is the most impacted factor with an increased expression of 3.14-fold. Other genes have the same behavior; the transcription factors *mtfA, atfA*, and *msnA* are over-expressed by 1.64, 1.42, and 1.67 folds, respectively. MAE also induces a significant downregulation of the Cu, Zn superoxide dismutase *sod1*, which is repressed by 1.62-fold. The expression of the other genes investigated (*cat2*, *mnSOD*, *atfB*, *ap-1*, and *srrA*) is not significantly affected. These results indicate that MAE targets several stress-signaling genes and in particular the transcription factors *mtfA*, *atfA*, and *msnA*, which are upregulated by MAE, thereby inhibiting AFB1 production. 

Concerning the global regulator m*tfA*, its overexpression has already been linked to decreased levels of *aflR* and thus to AFB1 inhibition in *A. flavus* [42]. Although *atf*A is involved in stress tolerance in several *Aspergilli*, it is also responsible for the expression of *catA* [43,44], which is consistent with the present results. The expression of *catA* is also reportedly governed by the bZIP transcription factor *atfB*, although the present study detects no significant changes in the latter. 

Finally, the expression of *msnA*, a gene required to maintain the normal oxidative stress status in *A. flavus* and *A. parasiticus*, is also enhanced. The deletion of *msnA* in both species leads to the accumulation of AFB1 and reactive oxygen species [45]. Moreover, the expression of genes encoding for catalases and superoxide dismutases in *A. flavus* and *A. parasiticus* are also regulated by *msnA* [45]. Thus, in the present study, the enhanced levels of *catA* expression may be partially attributed to the over-expression of *atfA* or to *msnA* levels.

Note that previous studies also reported the over-expression of the transcription factors *msnA* and *mtfA* when AFB1 is inhibited by *M. graeca* and eugenol treatment in *A. flavus* [19,46].

Thus, in the present study, AFB1 production is transcriptionally inhibited by MAE in a process accompanied by an over-expression of the transcription factors involved in fungal stress response (e.g., *mtfA, atfA*, and *msnA*). Moreover, the genes *catA* and *sod1* involved in the *A. flavus* anti-oxidant system are also affected by the treatment. 

### 2.2. Fractionation and Characterization of Fractions

To better identify the nature of the anti-AFB1 compounds in MAE, the extract was characterized and fractionated to analyze the activity of the different fractions against AFB1.

#### 2.2.1. Composition

*M. tenuiflora* bark was used for all the studies reported herein. It contains 88% DM and 12% water and volatile compounds. This dry bark is composed of roughly 50% fibers (29.6% cellulose, 8.7% hemicellulose, and 12.2% lignin), 10% proteins, and 2.3% minerals. Table 1 recapitulates the composition of MAE and the subsequent fractions. 

After 15 h, the aqueous extraction yields 11% of total dry plant mass, which is lower than previously reported for an ethanol extract of *M. tenuiflora* bark (16%) [31] and for a methanol extract macerated for 144 h (22%) [47]. However, the yield exceeds the 1% yield obtained for a hydroethanol extract of the *M. tenuiflora* bark after maceration for 5 days [33]. These results confirm that extraction conditions affect the results, particularly the solvent polarity but also the temperature and duration. In addition, the variability of the vegetal matter may also contribute to these different yields.

MAE has a total polyphenol content (TPC) of 397 mg gallic acid equivalent (GAE) per gram of DM and 172 mg/g DM of condensed tannins. This TPC is over twofold greater than that previously reported for a hydroethanolic extract of *M. tenuiflora* bark (156 mg GAE/g DM) [33] and falls in the same range as that obtained from ethanolic maceration of *M. tenuiflora* bark (360 mg GAE/g DM) [27].

This TPC also exceeds that obtained from other bark aqueous extracts obtained by various extraction methods (stirring, sonication, microwave-assisted extraction): *Fagus sylvatica* L. (29–57 mg GAE/g DM), *Eucalyptus globulus* (181 mg GAE/g DM), *Quercus* (15 mg GAE/g DM) [48], and *Acacia catechu* (131 mg GAE/g DM) [49].

Condensed tannins are high-molecular-weight polyphenols and are a major compound in barks. They are known to play a role in plant defense systems and to have various biological activities, such as herbivore repellant, antibacterial activity, and antifungal activity [50]. 

MAE contains 172 mg/g DM of condensed tannins, corresponding to a large proportion of TPC. For comparison, Pycnogenol^®^, which is pine maritimus bark extract (known as a reference for high content of condensed tannins), obtained through the Masquelier procedure, was titrated at 700 mg/g DM when extracted under optimal conditions (i.e., a high ethanol: water ratio of 70:30) [51]. 

To better understand how polyphenols and condensed tannins affect the fungal response (i.e., the growth and AFB1 production), we applied a bio-guided fractionation. The overall fractionation process comprised two successive steps: macroporous absorption resin (MAR) purification leading to fraction F (with a 74% yield) and polyvinylpolypyrrolidone (PVPP) purification leading to fraction Sf, corresponding to 8% of the MAE. MAR purification was chosen to refine polyphenols, allowing the separation of other hydrophilic compounds (saccharides, organic acids). The following PVPP fractionation isolated non-tannic compounds from precipitated tannins. The large quantity of condensed tannins in the MAE leads to an exceptionally-low-concentration fraction Sf having only 0.1% of DM (see Table 1).

The MAR step produces a gain in TPC of up to 551 mg GAE/g DM in fraction F. After the second step, Sf, corresponding to only 28% of F, contains 96 mg GAE/g DM polyphenols, which mostly corresponds to non-tannic polyphenols. As expected, Sf displays almost no condensed tannins (Figure 4) and much less TPC (85% less compared with F).

In parallel, the anti-oxidant activity of MAE and subsequent fractions was evaluated by using the 2,2-diphenyl-1-picrylhydrazyl (DPPH) free radical. MAE displays a high anti-oxidative activity, with an IC50 of 10 mg/L. By comparison, Trolox^®^, which is commonly used as anti-oxidant reference, has an IC50 of 3.5 mg/L [52]. This result is consistent with the results of Nascimento et al. [33], who obtained an IC50 of about 7 mg/L for hydroalcoholic fractions from *M. tenuiflora* bark. The anti-oxidant activity increases in F (with IC50 varying from 10 to 9 mg/L) due to its higher content of polyphenols but decreases in Sf (IC50 > 400 mg/L) due to the loss of condensed tannins. These results provide additional evidence of the strong anti-oxidant activity responsible for the various biological activities of condensed tannins [53]. 

#### 2.2.2. Characterization by High-Performance Liquid Chromatography

To better characterize how successive fractionations affect the MAE composition, we used high-performance liquid chromatography UV diode-array detection (HPLC-DAD) to analyze the MAE, F, and Sf. Figure 4 compares the F and Sf chromatograms at 280 nm.

A large peak appears between minutes 8 and 24 in the F chromatogram, which, based on the UV absorption spectrum of tannins, is attributed to tannin absorption [54]. The pattern also appears clearly in the MAE chromatogram (data not shown). The diversity of these oligomeric compounds generates a wide molecular-weight dispersity manifested by non-separated peaks. Moreover, this specific pattern does not appear in the Sf chromatogram after PVPP fractionation (see red curve in Figure 4), confirming the elimination of condensed tannins.

After 26 min, the F and Sf fractions produce essentially the same chromatograms, which is consistent with the presence of small-molecular-weight phenolic compounds in both extracts, only differing by their relative proportions. These compounds may correspond to flavonoids given their specific UV absorption spectra. Note that flavonoids have already been detected in *Mimosa tenuiflora* bark [31].

### 2.3. Effect of Fractions on Fungal Growth and Aflatoxin B1 Production

#### 2.3.1. Effect of Fractions on Fungal Growth

Fungal growth in five different concentrations of F and Sf (0.025, 0.05, 0.10, 0.15, and 0.20 mg DM/mL) were compared with that in a control culture without extract. After 8 days of incubation for both the control and non-control samples, a slight but statistically significant reduction in growth appears for concentrations above 0.025 mg of DM/mL, with the maximum inhibition for the fungus incubated with 0.2 mg of DM/mL being 5.74% and 3.95% for F and Sf, respectively. Thus, fractionation does not significantly modify the antifungal activity of MAE.

#### 2.3.2. Decreased Production of Aflatoxin B1

To evaluate how phenolic compounds and condensed tannins lead to the anti-aflatoxin activity of MAE, we tested how both F and Sf affect toxin production. Figure 5 shows the results.

The anti-AFB1 activity of F exceeds that of the MAE, with an IC50 of 0.042 mg DM/mL versus 0.15 mg DM/mL for MAE. This may be related to the greater proportion of polyphenols in F than in MAE. In contrast, the anti-AFB1 activity of Sf (IC50 of 0.238 mg DM/mL) is strongly reduced compared with that of MAE. Since this latter contains few condensed tannins, these results suggest that these compounds could be responsible for most of the anti-aflatoxin activity of MAE and F. To better characterize the possible role of condensed tannins in the reduction by MAE and F of AFB1 production, we now study how pure tannins affect toxin production.

### 2.4. Effect of Condensed Tannins on Fungal Growth and Aflatoxin B1 Production

Five different concentrations (0.025, 0.05, 0.10, 0.15, and 0.20 mg DM/mL) of experimental pine bark condensed tannins (PBCT) were tested to determine how they affect both *A. flavus* growth and AFB1 production.

Eight days of incubation causes a statistically significant reduction in growth for concentrations greater than 0.10 mg of DM/mL. The maximum inhibition is 9.5% after incubation with 0.8 mg condensed tannins/mL. 

Figure 6 shows that an increased concentration of condensed tannins (PBCT) inhibits AFB1 production. Over 50% inhibition of AFB1 production compared with the untreated control sample appears after incubation with 0.20 mg tannins/mL. By comparison, at 0.10 and 0.20 mg DM/mL, F produces a similar or even mildly stronger effect on AFB1 production (Figure 6), confirming the importance of condensed tannins in the anti-AFB1 activity of MAE. The effect of tannins on AFB1 production was first described in 1990 by Azaizeh et al. [55] when they demonstrated that water-soluble tannins contained in kernel coats and cotyledon of peanuts strongly reduce both *A. parasiticus* growth and aflatoxin production. It was also demonstrated that detannin-caffeinated coffee is significantly more sensitive to aflatoxin contamination than regular coffee [56]. More recently, it has also been demonstrated that some condensed tannins (proanthocyanidins) could alleviate AFB1-induced oxidative stress and apoptosis in broilers [57]. In fact, AFB1 and some of its metabolic precursors, such as norsolorinic acid and versicolorins, are highly oxygenated molecules and thus are subjected to redox regulation [58]. It has even been suggested that AFB1 production could be, by itself, part of the fungal response to oxidative stress because AFB1 biosynthesis is activated by high levels of oxidative stress [39]. The potent anti-oxidative activity of tannins present in MAE could therefore lead to a down-modulation of AFB1 synthesis.

## 3. Conclusions

This study discusses the anti-AFB1 activity of *Mimosa tenuiflora* bark aqueous extract (MAE). The inhibition of AFB1 production is related to a transcriptional inhibition caused by a downregulation of the genes involved in the biosynthetic pathway, and especially of the two internal regulators *aflR* and *aflS*. Moreover, in *A. flavus*, this response is accompanied by a modulation of the gene coding for the fungal stress response and the anti-oxidant defense system. A bio-guided fractionation of MAE strongly suggests that condensed tannins play a key role in this anti-AFB1 activity related to their strong anti-oxidative capacity. These results allow considering the use of such plant extracts to protect crops from aflatoxin B1 contamination. The interest of an aqueous extract is that it could be used directly in the fields using irrigation systems. Its use on grains during storage will require finding an adequate dry formulation.

## 4. Materials and Methods

### 4.1. Materials

#### 4.1.1. Solvents and Standards

All solvents (hydrochloric acid; Tween 80; chloroform; acetonitrile; formic acid and methanol) and chemicals (polyvinylpolypyrrolidone; Folin-Ciocalteu reagent; DPPH reagent; AFB1 standard) were purchased from Sigma–Aldrich (St Quentin Fallavier, France) with the exception of ethanol 96% and sodium carbonate purchased from VWR International (Fontenay sous bois, France). The solvents used in this study were HPLC grade. Ultrapure water was prepared using Veolia Purelab Classic (Veolia, Toulouse, France).

#### 4.1.2. Plant Materials

The stem bark of *M. tenuiflora* (Willd.) was purchased from Red Mexicana de Plantas Medicinales y Aromaticas (REDMEXPLAM) and was botanically identified with the registration number UATX/01/Tepezcohuite/2019. The botanical sample was deposited at Jardin Botanico Universitario UAT. Dried stem bark was ground by using a mill equipped with a 1 mm grid, and the powder was stored in plastic bags at 4 °C until use.

### 4.2. Plant Characterization

Dry matter content was determined by weighing each sample before and after drying in a Memmert oven at 103 °C (Schwabach, Germany) until reaching a constant weight. Mineral content was determined by heating at 550 °C for 6 h. The protein content was determined by applying the Kjeldahl method, as described by Kjeldahl [59]. Nitrogen content was multiplied by 6.25 according to the French standard NF V18-100 (AFNOR). The fiber content (cellulose, hemicellulose, and lignin) was determined by using the ADF–NDF method of Van Soest and Wine with a Tecator Fibertec M1017 (FOSS) [60].

### 4.3. Preparation of MAE, F, and Sf

#### 4.3.1. Preparation of M. tenuiflora Aqueous Extract

A total of 120 g of ground *M. tenuiflora* bark was extracted with 4 L of distilled water under mechanical agitation for 15 h at room temperature. The extract was centrifuged in a sigma 6–16 k centrifuge at 15,000 relative centrifugal field for 15 min (Sigma Laborzentrifugen Gmbh, Osterode am Harz, Germany) and filtered through Whatman grade 1 filter paper (GE Healthcare Life Sciences, Vélizy-Villacoublay, France). Filtrates were adjusted to 120 mL with distilled water and sterilized in an autoclave at 121 °C for 20 min (SMI group UNICOM, Montpellier, France). The final sterile extract was stored at +4 °C until use. Stock filtrates were diluted to experimental concentrations (2 g/L) with distilled water.

#### 4.3.2. Fractionation of MAE

##### MAR Fractionation

MAE was fractionated by using microporous adsorption resins (MARs). Fifty grams of the FPX66 adsorbent resin (Rohm and Haas, Philadelphia, PA, USA) were preconditioned with ultrapure water (pH 5.5) and packed in a cylindrical glass column (ID × L = 3 × 30 cm) equipped with a fritted disk. Two hundred mL of the MAE at 3.245 g DM/L were loaded into the column. Elution was done first with 2BV (bed volume) of ultrapure water to remove sugars and other compounds, following which desorption was done using 1BV of 250 mL of 95% ethanol to release the compounds retained on the resin [61]. The whole process was repeated four times, and the ethanolic fractions were pooled to obtain the fraction (F). 

##### PVPP Fractionation

The fraction F (2.64 g/L) was treated with polyvinylpolypyrrolidone (PVPP) using the method of Peng et al. (2001) with slight modifications [62]. Five mL of each sample were mixed with 5 mL of water and 500 mg of PVPP. After vortexing for 30 s, the samples were maintained at 4 °C for 15 min and then stirred again for 30 s before being centrifuged at 3000× *g* for 10 min [63]. The supernatant was then collected for further analyses (subfraction Sf).

### 4.4. Characterization of MAE and Fractions F and Sf

#### 4.4.1. Dry Matter Content

The dry matter content of MAE, F, and Sf was determined as described for *Mimosa* bark.

#### 4.4.2. Total Phenolic Content

The total amount of phenols in MAE and in the fractions was determined by using the Folin-Ciocalteu method adapted from Singleton and Rossi [64]. Briefly, 20 µL of the sample were mixed with 10 µL of Folin-Ciocalteu reagent and 170 µL of sodium carbonate at 2.36% in a 96-well microplate. The absorbance was measured at 700 nm using a BMG-LabtechSpectrostar-Nano spectrophotometer (BMG LABTECH SARL, Champigny s/Marne, France) with a reaction time of 45 min at 45 °C. The results are expressed in milligrams of GAE per gram of dry extract.

#### 4.4.3. Condensed Tannin Content

The condensed tannin content of the samples was determined by using an adaptation of the Waterman and Mole method [65]. Samples were first diluted with water to obtain an absorbance of around 0.2 after hydrolysis. To each 2 mL sample was added 1 mL distilled water and 3 mL of concentrated hydrochloric acid (12 N) in one unheated test tube (control sample) and in one heated test tube. The latter was maintained at 100 °C for 30 min while the control tube was immersed for the same duration in crushed ice. The heated test tube was then recovered and cooled on crushed ice. A volume of 0.5 mL ethanol was added to each tube and the tubes were vortexed for 10 s. The 550 nm absorbance of the control and test samples was recorded by using a Shimadzu UV1800 spectrophotometer (Shimadzu Corp., Kyoto, Japan). The tannin concentration was calculated using
Tannin (mg/g) = [0.3866 × (absorbance_sample_ − absorbance_control_) × dilution factor]/DM.(1)

Hydrolysis was done in triplicate for each sample, and the results expressed as mg condensed tannin per g DM ± standard error of the mean.

#### 4.4.4. Radical-Scavenging Activity

The anti-oxidant activity of the extracts was determined by the antiradical activity by using the 2,2-diphenyl-1-picrylhydrazyl (DPPH) method adapted from Brand-Williams et al. [66]. Briefly, the DPPH reagent (150 µL) was mixed with seven different concentrations of the sample (150 µL), and the absorbance of the mixtures was measured at 516 nm after reacting for 40 min in a BMG-LabtechSpectrostar-Nanospectrophotometer (BMG LABTECH SARL, Champigny s/Marne, France). Measurements were made of five replicates of each sample concentration. The reaction equation is
*f*(*Ce*)*c* = 1 − *A*/*A*_0_,(2)
where *Ce* is the concentration of the extract (in g/L), *A* is the absorbance of the extract, and *A*_0_ is the absorbance of the blank (DPPH) after reacting for 40 min. The radical-scavenging activity is reported as the half-maximal inhibitory concentration (IC50), which is the concentration of the extract or fraction that inhibits 50% of the DPPH radicals, expressed in grams of extract per liter (g/L). The calculation of IC50 was done by using linear regression, as follows:IC50 = 0.5 − *a*/*b*,(3)
where IC50 is the half-maximal inhibitory concentration, *a* is the origin ordinate, and *b* is the slot.

### 4.5. Effect of M. tenuiflora Extract and Fractions on Aspergillus flavus Growth and Aflatoxin B1 Synthesis

#### 4.5.1. Fungal Strain and Culture Conditions

The *Aspergillus flavus* strain NRRL 62477 was used for all assays [67]. For all experiments, 1000 spores of *A. flavus* were inoculated centrally into the culture medium using 10 µL of a spore suspension prepared in Tween 80 from a 7-day-old culture (10^5^ spores/mL). The culture medium was composed of 18 mL of malt extract agar (Biokar Diagnostics, Allone, France) and 2 mL of autoclaved *M. tenuiflora* extract or fraction, prepared at five different concentrations by dilution in water. Control cultures were made by adding 2 mL of water to the initial 18 mL of malt extract agar. Cultures were incubated for 8 days at 27 °C. Each assay was done in triplicate. After incubation, the growth was quantified in terms of the measured colony diameter. 

#### 4.5.2. Extraction and Quantification of Aflatoxin B1 by HPLC

AFB1 analysis was done as described by El Khoury et al. [19]. In brief, AFB1 extraction was done with 30 mL of HPLC-grade absolute chloroform. Next, supernatants were filtered through a Whatman 1PS phase separator (GE Healthcare Life Sciences, Vélizy-Villacoublay, France). Two milliliters of filtrate were evaporated to dryness in a STUART SBH200D/3 sample concentrator (Stuart equipment, Paris, France) at 45 °C, and the samples were re-solubilized in 2 mL of acetonitrile. Finally, the samples were filtered through 0.45 μm disk filters (Thermo Fisher Scientific, Illkirch-Graffenstaden, France) and placed in HPLC vials. AFB1 was analyzed by using an Ultimate 3000 UPLC (Thermo Fisher Scientific, Illkirch-Graffenstaden, France) with an EvoC18 column (3 µm, 150 × 3.2, Phenomenex, Le Pecq, France) conditioned at 27 °C. The elution program used for separation consisted of forming an isocratic mixture composed of acetonitrile and water (25:75 *v*:*v*). The mobile phase had a flow rate of 1.2 mL/min. Ten microliters of sample were injected. AFB1 was detected by using a fluorescent detector at 365 (430) nm excitation (emission) wavelengths. The identity of the molecule was confirmed by analyzing the UV absorption spectrum by an additional diode matrix detector (DAD) coupled to the system. AFB1 production levels were calculated based on a standard calibration curve (0.16 to 20 mg/L).

### 4.6. Effect of M. tenuiflora Extract on Gene Expression 

#### 4.6.1. Fungal RNA Isolation and Reverse Transcriptase-Polymerase Chain Reaction 

The culture conditions for analyzing gene expression were the same as described in Section 4.5.1. A concentration of 18.87 mg/mL of the *M. tenuiflora* extract was chosen and cultures were incubated for 4 days at 27 °C. Sterile cellophane films were used to separate fungal mycelia from media and six biological replicates of each condition were prepared. Fungal RNA was obtained by grounding mycelia under liquid nitrogen and further purified by using a Qiagen RNeasy Plus Minikit (Qiagen, Hilden, Germany) according to the manufacturer instructions. The quality and quantity of fungal RNA were respectively determined by gel electrophoresis (1.2% of agarose) and NanoDrop ND1000 (Labtech, Palaiseau, France). A concentration of 300 ng/µL was determined for all analyses. First-strand cDNA synthesis was done by reverse transcriptase polymerase chain reaction (RT-PCR). We used a 20 µL mixture containing 20 U of RNase inhibitor (Thermo Scientific, Illkirch, France), 4 μL of 5 × reaction buffer, 200 U of RevertAid reverse transcriptase, 2 μL of 10 mMdNTP (Euromedex, Souffelweyersheim, France), 1 μL sterile water, 1 μL oligo (dT) Bys 3′ Primer: (5′-GCTGTCAACGATACGCTATAACGGCATGACAGTGTTTTTTTTTTTTTTT-3′), and 10 µL of RNA. Reverse transcription conditions were the same as in previous work [46].

#### 4.6.2. q-PCR Conditions and Analysis of Gene Expression 

All primer sets have been described previously [46]. The choice of primers used in this study was based on their relation with AFB1 production and the oxidative stress response in *A. flavus.* Experiments were performed in 384-well plates prepared by an Agilent Bravo Automated Liquid Handling Platform (Agilent Technologies, Santa Clara, CA, USA). Each plate-well contained 2.5 μL of Power SYBR^®^ Green PCR Master Mix (AppliedBiosystems, Warrington, UK), 1.5 μL of each primer set, and 1 μL of cDNA material. The *q*-PCR experiments were done by using a ViiA7 Real-Time PCR System (AppliedBiosystems, Forster City, CA, USA) under conditions adapted from previous work [46]. Gene expression was analyzed by using Quant-Studio Real Time PCR software v1.1 (Applied Biosystems, Courtaboeuf, France), and differences between conditions (control versus treatment) were determined by using the 2^−ΔΔCt^ analysis method [68]. Data were normalized by using the housekeeping gene β-tubulin that was proved to be the most stable reference after analysis with the Normfinder algorithm [69]. All graphics show normalized values, with the control levels set to unity.

### 4.7. Characterization of Compounds Present in M. tenuiflora Extract and Fractions

The HPLC-UV (DAD) analysis was done by using an Ultimate 3000 UPLC (Thermo Fisher, France), which consists of a vacuum degasser, a quaternary pump, an automatic sampler, a temperature-controlled chamber, and a photodiode array detector (DAD), all piloted by a Chromelon chromatography data system (Thermo Fisher, France). The analyses were done in the reverse phase on an F5 column (3 µm, 150 × 3.0 mm, Phenomenex, Le Pecq, France). The mobile phase was composed of solvents A and B, which are respectively water with 0.1% formic acid and methanol with 0.1% formic acid. The elution gradient was: 5% (B) in 5 min, 5% to 95% (B) in 45 min, 95% (B) in 5 min, 95% to 5% (B) in 5 min, and initialization at 5% (B) in 5 min. The flow rate was 0.3 mL/min. The analysis was done at 35 °C. The injection volume of the samples was 20 μL. The solutions to be analyzed were prepared at 2 mg/mL. DAD acquisitions were made in the range 190–450 nm.

### 4.8. Statistics

The data were analyzed by using a one-way analysis of variance test to determine the differences between the control and treated groups. Differences were considered statistically significant for *p*-values less than 0.05. All data errors are presented as the standard error of the mean.

## Figures and Tables

**Figure 1 toxins-13-00391-f001:**
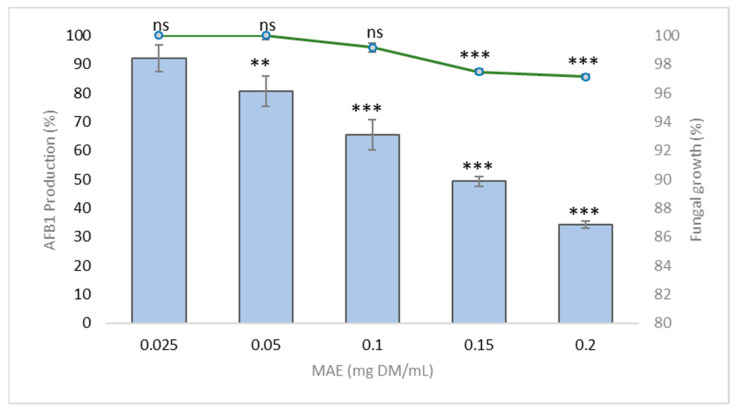
AFB1 production (blue bars) and fungal growth (green line) in *A. flavus* NRRL 62477 strain in various concentrations of MAE. Results are expressed as percentage of untreated control ± standard error of the mean (*n* = 3). ns = no statistically significant change; ** *p*-value < 0.01; *** *p*-value < 0.001.

**Figure 2 toxins-13-00391-f002:**
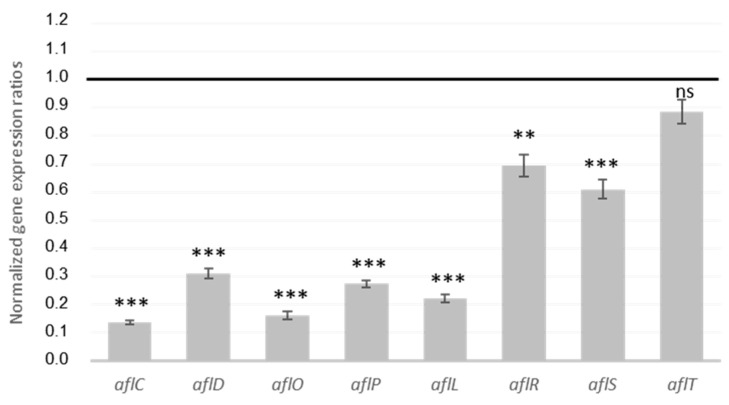
Normalized gene expression ratio for genes from the AFB1 cluster in culture medium with *Mimosa* Aqueous Extract (MAE). Black line indicates level of expression in control medium (i.e., without MAE). ns = no statistically significant change; ** *p*-value < 0.01; *** *p*-value < 0.001.

**Figure 3 toxins-13-00391-f003:**
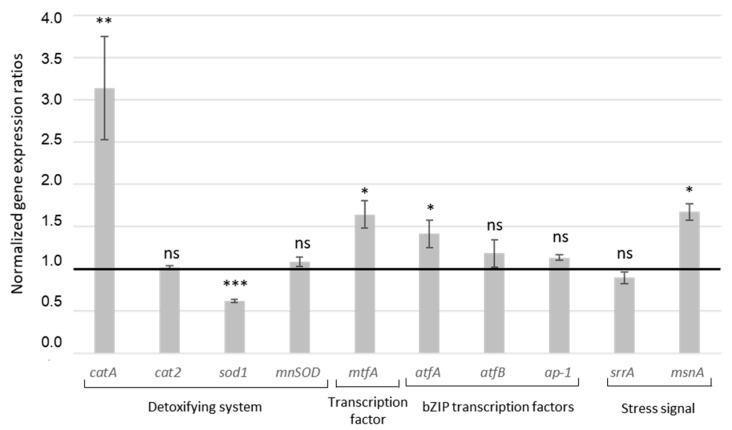
Normalized gene expression ratio for genes involved in oxidative stress response in *A. flavus* and exposed to MAE. Black line represents expression levels of genes in control sample. ns = no statistically significant change; * *p*-value < 0.05; ** *p*-value < 0.01; *** *p*-value < 0.001.

**Figure 4 toxins-13-00391-f004:**
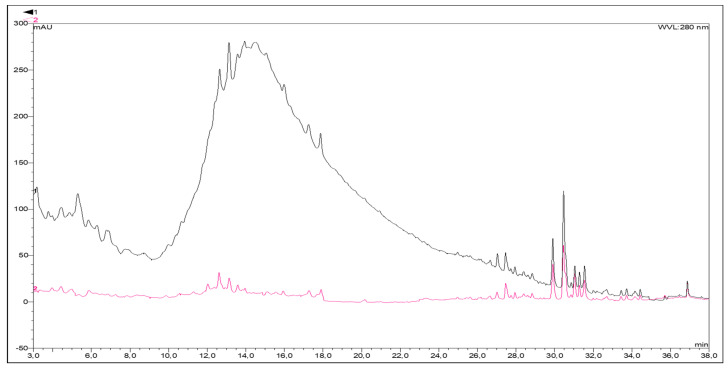
Results of high-performance liquid chromatography UV diode-array detection (HPLC-DAD) at 280 nm of fraction F (black curve) and Sf (red curve) at a concentration of 2 g/L. Compounds were separated in a Kinetex F5 analytical column; mobile phase was methanol and water in 0.1% formic acid, gradient elution, at 35 °C. Chromatographic conditions are reported in Section 4.7.

**Figure 5 toxins-13-00391-f005:**
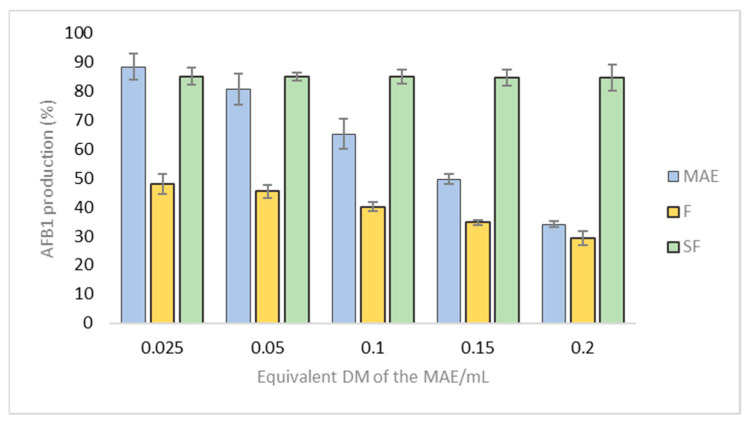
AFB1 production for F (yellow bars) and Sf (green bars) compared with the same for MAE (blue bars). Results are presented in terms of equivalent DM of MAE/Petri dish and expressed as the percentage (%) of AFB1 production compared with untreated control cultures. Histogram shows mean ± standard error of the mean for four experiments.

**Figure 6 toxins-13-00391-f006:**
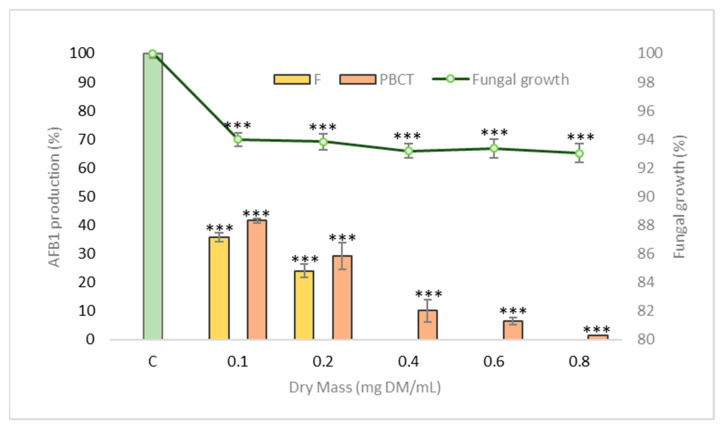
AFB1 production and fungal growth (green line) of *A. flavus* NRRL 62477 strain when exposed to PBCT (pine bark extract, orange bars) compared with exposure to 0.1 and 0.2 mg DM/mL of F (yellow bars). F was not tested at higher concentrations (0.4, 0.6 and 0.8 mg/mL). Results are expressed as a percent of AFB1 production of untreated control ± standard error of the mean (*n* = 4). *** *p*-value < 0.001.

**Table 1 toxins-13-00391-t001:** Characterization of MAE, F, and Sf.

Purification Step	Mass Proportion of Total Dry Plant (%)	Fraction Step Yield (%DM)	Polyphenols (mg GAE/g DM Extract) ^1^	Condensed Tannins (mg/g DM)	Anti-Oxidant Activity on DPPH IC50 (mg/L) ^2^
Aqueous Extract (MAE)	11.15	11	397 ± 22	171.6 ± 2.8	10
Fraction (F)	1.82	74	551 ± 11	332.3 ± 0.6	9
Subfraction (Sf)	0.10	28	96 ± 2	2.8 ± 0.2	>400 ^3^

^1^ GAE: gallic acid equivalent. ^2^ DDPH: 2,2-diphenyl-1-picrylhydrazyl. Results expressed in terms of IC50, which is the concentration of the extract or the fraction that reduces DPPH radicals by 50%, expressed in milligrams of extract per liter (mg/L). ^3^ over 400.

## Data Availability

The data presented in this study are available on request from the corresponding author.

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
