# Peer review of "Mimosa tenuiflora Aqueous Extract: Role of Condensed Tannins in Anti-Aflatoxin B1 Activity in Aspergillus flavus"

_toxins, 2021, doi:10.3390/toxins13060391_

Round 1
Reviewer 1 Report
Overall Comments:
This manuscript described that using aqueous extract of Mimosa tenuiflora bark affects both the growth of Aspergillus flavus and AFB1 production. This manuscript is not well written and several points need to be clarified. Such as using the aqueous extract of the Mimosa tenuiflora bark in the experiment. There are so many compounds inside the aqueous extract mixture. It can not tell which components are key compounds for the inhibition of Aspergillus flavus growth. Therefore, I am not sure which polyphenol or tannin are you looking for in the HPLC analysis. The unknow peak in 8-26 min are mixture of several compounds which have an UV absorbance at 280 nm. Overall, this manuscript is not qualified for the Toxin.
Specific comments:
P3 line 113.. The two main internal regulators are down-modulated with fold changes of 1.4 for aflR and 1.6 for aflS compared with the control. Moreover, gene coding for stable AF-intermediates intervening at the first (aflC, aflD), middle (aflL), and late (aflO, aflP) stages of the enzymatic cascade is also hindered by MAE. In contrast, aflT is not significantly affected, which is consistent with the fact that this gene is regulated by neither aflR nor aflS. -->Do you mean that the fold changes of 1.4 for aflR 114 and 1.6 for aflS are not important?
P4 Figure 2. Please show the mRNA level and protein level for AFB1 synthesis cluster gene.
P5 Figure 3 Please show the mRNA level and protein level of genes involved in oxidative response.
P7 Figure 4. There is no standard compound in the HPLC assay. I am not sure which compounds are you interested. The UV-280 nm may detect so many unknow proteins but not the phenol or tannin.
P9. Line 330 .. Solvents and Standards--> Please state the chemicals and equippment that you used.
Author Response
Please see the attachement

Reviewer 2 Report
The manuscript describes the inhibition of aflatoxin production by tannins derived from aqueous extraction of Mimosa tenuiflora bark. The authors analyzed gene expression in Aspergillus flavus in the presence of the extract to illuminate the fungal response mechanisms and determined that two aflatoxin biosynthesis transcription factors are affected, as well as some stress-related genes. In addition, the authors conducted chemical analyses to characterize the extract’s chemical composition and properties. They also purified the extract to narrow down the bioactive constituents. Figures 4 and 5 and suggest that the tannin fraction are responsible for the observed inhibition. The manuscript is well thought out, well written, and easy to follow. The conclusions are justified by the data provided. Please address some minor points before publication:
Line 188: “After 15 hours, the aqueous extraction yields 11%”... of? Mass of extract in proportion of total dry plant mass? Please clarify in text. Table is clear.
Clarify preparation of the MAE and dilutions. If I’m understanding correctly, consider changing Line 352 to “Stock filtrates (30 g/L) were diluted to experimental concentrations with distilled water and sterilized…”
Line 314: figure 6 legend. Consider adding “F was not tested at higher concentrations: 0.4, 0.6, 0.8 mg/ml”
Line 332 Plant materials: Was stem bark or root bark used in this study? The authors are likely aware that the root bark has a high DMT content and is used as an entheogen. Plant material containing DMT seem to skirt the legal system, but DMT itself is a controlled substance in the US.
Conclusion: Can the authors elaborate on how this extract or information gleaned in this study can be used practically to prevent aflatoxin? Would a solution of the tannins be sprayed directly on crops? Pre or post harvest? Would appreciable levels of DMT be carried in to the food supply?
Author Response
please see attachement

Reviewer 3 Report
Introduction: Different strategies to reduce AFB1 contamination are briefly discussed but no mention is made of biological control methods. Biological control using atoxigenic isolates of Aspergillus flavus have been proven to be effective to reduce AFB1 levels to the extent safe for human and animal consumption. Check these and many other articles; https://doi.org/10.3390/agronomy10040491; https://doi.org/10.3920/WMJ2016.2130
Results: 2.1.2. Reduced production of AFB1
MAE is shown to reduce AFB1 production by 50% as compared to the untreated control. But the AFB1 concentration of untreated control is never mentioned. It is somehow misleading to say that MAE reduces AFB1 by a given percentage without mentioning what is the AFB1 concentration in the control. Measurement given in parts per million or any other absolute measure would be easier to make the comparison.
Results: 2.1. 3. MAE inhibition of genes from AFB1 cluster
Measurement of gene expression analysis shows that MAE can have modest inhibition of some of the key genes of aflatoxin gene cluster. But what hasn't been shown is that the reduction is due to MAE but not because of other factors. AFB1 production is influenced by a lot of environmental factors and the level of reduction seen in the result could also be influenced by those environmental factors. Furthermore, the reduction of gene expression level of aflR, aflS, and aflT is very modest and may not have any impact on overall AFB1 production.
Results. 2.1.4. Effect of MAE on Expression of Genes Involved in Oxidative Stress Response
The oxidative stress-related genes tested for gene expression level are very common genes that can be influenced by any stress factor. Without controlling other environmental factors, just looking at their expression level with response to MAE application is not very informative. Furthermore, very modest changes in the expression of those genes possibly suggest that the effect may not be very significant.
Results. 2.3. Effect of Fractions on Fungal Growth and AFB1 Production
Looking at the results, the application of MAE seems to have an effect on AFB1 production but no such effect is seen on fungal growth. I was wondering if the reduction in AFB1 production is a transient phenomenon due to the slowed growth of the fungi. Would it be possible that once the effect of MAE (tannins and other compounds) subsidize, the AFB1 level would go higher? Different sets of experiments may be needed to answer these questions but what I am trying to understand is if the effect of tannins like compound is a temporary phenomenon or an event with a long-lasting effect.
Author Response
please see attchement

Round 2
Reviewer 1 Report
Therefore, I am not sure which polyphenol or tannin are you looking for in the HPLC analysis. The unknow peak in 8-26 min are mixture of several compounds which have an UV absorbance at 280 nm.
Answer: These compounds are condensed tannins as demonstrated by the chromatogram obtained after PVPP precipitation in which this large peak is no more present. Moreover, these molecules can’t be proteins since they were removed from the extract by column separation done on Amberlite resin.
This answer can not convice the reviewer. How do you know the Amberlite can absorb the whole proteins ??
